# Increase in Lactulose Content in a Hot-Alkaline-Based System through Fermentation with a Selected Lactic Acid Bacteria Strain Followed by the β-Galactosidase Catalysis Process

**DOI:** 10.3390/foods12234317

**Published:** 2023-11-29

**Authors:** Yaozu Guo, Wenlong Ma, Manxi Song, Wenqiong Wang, Boxing Yin, Ruixia Gu

**Affiliations:** 1College of Food Science and Engineering, Yangzhou University, Yangzhou 225127, China; gyz15240477626@163.com (Y.G.); wenlong-ma@yzu.edu.cn (W.M.); smx950101@outlook.com (M.S.); rxgu@yzu.edu.cn (R.G.); 2Jiangsu Dairy Biotechnology Engineering Research Center, Kang Yuan Dairy Co., Ltd., Yangzhou University, Yangzhou 225127, China

**Keywords:** lactulose preparation, lactic acid bacteria, chemical synthesis, enzymatic catalysis

## Abstract

In this study, lactic acid bacteria (LAB) fermentation and β-galactosidase catalysis methods were combined to increase the lactulose concentration and reduce the galactose and lactose content in a hot-alkaline-based system. The optimal conditions for chemical isomerization were 70 °C for 50 min for lactulose production, in which the concentration of lactulose was 31.3 ± 1.2%. Then, the selection and identification of LAB, which can utilize lactose and cannot affect lactulose content, were determined from 451 strains in the laboratory. It was found that *Lactobacillus salivarius* TM-2–8 had weak lactulose utilization and more robust lactose utilization. *Lactobacillus rhamnosus* grx.21 was weak in terms of lactulose utilization and strong in terms of galactose utilization. These two strains fermented the chemical isomerization system of lactulose to reduce the content of lactose and galactose. The results showed that the lactose concentration was 48.96 ± 2.92 g/L and the lactulose concentration was 59.73 ± 1. 8 g/L for fermentation lasting 18 h. The β-galactosidase was used to increase the content of lactulose in the fermented system at this time. The highest concentration of 74.89 ± 1.68 g/L lactulose was obtained at an enzymatic concentration of 3 U/mL and catalyzed at 50 °C for 3 h by β-galactosidase.

## 1. Introduction

Lactulose(4-O-β-D-gal-actopyranosyl-D-fructofuranose) is composed of galactose and fructose linked by β-1,4 glycosidic bonds and has received increasing attention in recent years due to its unique health-promoting functions. Lactulose is non-digestible, allowing water and electrolytes to be retained in the intestinal lumen, further increasing the osmotic pressure in the intestines, thus producing an antidiarrheal effect and relieving chronic constipation [1]. Due to its excellent efficacy and safety profile, the total use of high-purity lactulose has increased annually [2].

There are two ways to synthesize lactulose: chemical and biological. The chemical synthesis of lactulose mainly involves catalyst isomerization and electrochemistry isomerization in two ways. At present, the yield of lactulose synthesized by the chemical method is about 75–85%, and the molar ratio of boric acid to lactose at this time is about 1:1. However, the addition of sodium aluminate, sodium hydroxide, and boric acid will pollute the environment and waste resources. The subsequent removal of aluminum, boron, and desalination is difficult [3].

At present, an innovative anionic extraction-assisted isomerization strategy, based on selective molecular recognition with boronates, is always used for producing high-yield lactulose. But inevitably, this approach must use organic extraction [4]. Agustina Fara et al. (2020) showed the different carbohydrate consumption potentials of lactic acid bacteria (LAB) strains. The use of LAB could not only purify lactulose of the chemical synthesis system of lactulose but also make it possible to add more nutrient (e.g., lactic acid) production to it [5]. Therefore, applying LAB in the selective separation and reduction of non-target products of the chemical synthesis of lactulose is a worthwhile and profound study.

Biological methods to synthesize lactulose are enzymatic and whole-cell catalysis. The enzymes used to produce lactulose are β-galactosidase and cellobiose differential isomerase. Vaheri et al. (1978) discovered for the first time that β-galactosidase can catalyze lactulose production from lactose and fructose, which started the enzyme-catalyzed preparation of lactulose [6]. β-galactosidase (EC 3.2.1.23) catalyzes the production of lactose from lactulose by using the hydrolytic activity of β-galactosidase to catalyze the production of galactose and glucose from lactose and the transglycosylation activity of β-galactosidase to transfer the galactosyl group to the fructose receptor to produce lactulose. During the reaction, β-galactosidase can also transfer galactoside to other hydroxyl-containing nucleophilic receptors, such as water and lactulose to generate different polymers, and these products in turn can be slowly hydrolyzed by the enzyme as substrates. Currently, the conversion of lactulose prepared by this method has been increased to between 42–60%. Cellobiose 2-epi-merase (EC5.1.3.1.1) catalyzes the differential isomerization of the D-glucosyl group at the reducing end of β-1,4-glycosidically linked oligosaccharides to D-fructose, i.e., it catalyzes the conversion of lactose to lactulose, and it can also differentially isomerize lactose to obtain lactose isomers, such as epsilon lactose [7].

During the enzymatic preparation of the lactulose reaction process, oligosaccharides are continuously generated and degraded in a dynamic equilibrium, and their content constantly changes in all reaction periods. The generation of lactulose requires a specific concentration of galactoside donor and acceptor fructose. Using the highest possible starting concentration of lactose can improve the conversion rate of lactulose [8]. Mingming Wang et al. (2015) used ethanol permeability treated *Escherichia coli* cells containing (cellobiose differential isomerase) as a whole-cell catalyst, and the reaction was carried out for 2 h at pH 7.5 and 80 °C, and the conversion rate of lactulose was 65%, which is the highest reported in the literature so far [9]. Yeast, *Bacillus subtilis* and *Escherichia coli*, is the commonly used host cell, and the use of Lactobacillus in lactulose preparation has not been reported. Lactobacilli, one of the most widely used probiotics, have an excellent immunomodulatory effect, regulating the body’s immunity and improving the intestinal barrier to enhance human immunity. It has been reported that lactic acid bacteria contain β-galactosidase and have transglycosidic activity, indicating that lactic acid bacteria have the potential to synthesize lactulose [5,10]. Enzymes extracted from lactic acid bacteria can be used in various food-related applications without extensive purification. The preparation of lactulose using probiotic lactic acid bacteria as the host to prepare lactulose-rich synbiotics will become a new development direction.

This paper is based on the screening of lactic acid bacteria (LAB) that can efficiently utilize lactose and galactose and have a weak ability to use lactulose to reduce the non-target product in the chemical synthesis system of lactulose such as lactose and galactose. The first step was the formation of lactulose by isomerizing the glucose moiety of lactose to a fructose moiety using a chemically catalyzed isomerization method in which a Lobry de Bruyn–van Ekenstein transformation occurs in a hot-alkaline-based environment. The extended reaction time of this single hot-alkaline-based catalysis preparation of lactulose from lactose is accompanied by a decrease in pH with the generation of by-products, which would reduce the possibility of the intramolecular recombination and isomerization of lactose to lactulose, followed by termination of the reaction at a specific time using a weak acid. Secondly, LAB were used to ferment the chemical synthesis solution of lactulose to raise the purities of lactulose and reduce the content of lactose and galactose. Finally, adding fructose and β-galactosidase improved the synthesis capacity of lactulose through a trans-glycosylation reaction [11]. This was to increase the target product of lactulose concentration in the fermented system. Therefore, the chemical isomerization, lactic acid bacteria (LAB) fermentation, and β-galactosidase catalysis methods were combined to increase the lactulose concentration and reduce non-target products of galactose and lactose content. This study can make preparing lactulose more compatible with green chemistry principles and sustainability.

## 2. Materials and Methods

### 2.1. Materials

Standards (lactulose, lactose, fructose, and galactose) were purchased from Beijing Solarbio Science & Technology Co., Ltd (Beijing, China). β-galactosidase, lactulose, and fructose were purchased from Shanghai yuanye Bio-Technology Co., Ltd. (Shanghai, China). Lactose was purchased from Shanghai Aladdin Biochemical Technology Co., Ltd. (Shanghai, China). Glucose and sodium hydroxide were purchased from Sinopharm Chemical Reagent Co., Ltd. (Shanghai, China). Galactose was from Shanghai Macklin Biochemical Co., Ltd. (Shanghai, China). Culture medium MRS broth and MRS (without glucose) were purchased from Qingdao Hi-Tech Industrial Park Haibo Biotechnology Co. (Qingdao, China). HPLC-grade acetonitrile was purchased. The experimental water was sterile ultrapure water.

In this study, 451 lactic acid bacteria (LAB) strains were obtained from the Key Lab of Dairy Biotechnology and Safety Control, Yangzhou University.

### 2.2. Culture Medium Preparation

The second generation of activated strains were obtained and incubated at 37 °C for 16 h in MRS medium for the following experiment. The weight of the MRS without glucose, 32.24 g, was added to sterile water to 1 L, which was set as the MRS–glucose (MRS-Glc) medium.

Afterward, 10 g/L of lactose (Lac) was added into the MRS-Glc medium, which was set as MRS-Glc + Lac medium. Next, 10 g/L of lactulose (Lu) was added into the MRS-Glc medium, which was set as MRS-Glc + Lu medium. Later, 10 g/L of galactose (Gal) was added to the MRS-Glc medium, which was set as MRS-Glc + Gal medium. These culture mediums were used to strain the culture. The MRS-Glc medium with 5 g/L lactulose, 5 g/L lactose, and 5 g/L galactose added was set as 0.5% Lu + 0.5% Lac + 0.5% Gal. The medium was used for strain screening.

### 2.3. Sequence Analysis of the 16 S rRNA Gene

The DNA of the strain was extracted according to the instructions of the kit, and the extracted strain DNA was used as a template for PCR amplification of the 16s rDNA gene. (Ezup Column Wash Bacterial Genomic DNA Extraction Kit, Sangyo Bioengineering Technology Co. Shanghai, China) The primers used were 27F (AGAGTTTGATCCTGGCTCAG) and 1492R (GGTTACCTTGTTACGACTT) [12]. Amplification was performed using a 25 μL reaction system. The system consists of 1 μL of DNA template, 0.5 μL + 0.5 μL of primer, 10.5 μL of ddH_2_O, and 12.5 μL of 2 × TaqDNA polymerase.

The reaction parameters of PCR include five mins of denaturation at 95 °C, followed by 30 cycles of 95 °C for 10 s, 53 °C for 15 s, 72 °C for 90 s, and a final extension at 72 °C for 10 min. After the reaction, 5 μL of PCR amplification products was detected by 0.1% agarose gel electrophoresis, and the bands were observed using a gel imaging system [13]. The PCR amplification products were entrusted to Shanghai Bioengineering for sequencing, splicing, and homology comparison using BLAST on the NCBI website.

### 2.4. Chemical Synthesis of Lactulose

Approximately 200 g of lactose was added to 1 L of ultrapure water. Then, the solution was stirred well, which adjusted the pH to 11 or 12 with 12.75 mol/L NaOH [14]. The temperature of the water broth was adjusted to 60 °C and 70 °C. Then, it was reacted for 50, 70, and 90 min to take samples [11]. High-performance liquid chromatography (HPLC) measurements were carried out to determine the concentration of lactulose.

### 2.5. Enzymatic Synthesis of Lactulose

Use of the enzymatic catalysis system began after LAB fermentation. The fructose was added in the ratio of 2:1 (lactose: fructose *w*:*w*). The *E. coli*-derived β-galactosidase was added at 1 U/mL, 2 U/mL, 3 U/mL, and 4 U/mL and reacted in a water bath at an optimum temperature of 50 °C and an optimum pH of 7 for 18 h [15]. HPLC was used to test the content of lactulose and lactose in the supernatants of the samples at 0 h, 3 h, 6 h, 12 h, and 18 h.

### 2.6. Chromatographic Analysis of Carbohydrates

HPLC was used to determine the contents of lactulose, lactose, and galactose during fermentation. The chromatographic column was an All Chrom Rocksil Carbohydrate ES 5u (250 mm × 4.6 mm, 5 μm) from Agilent Technologies Inc. with a column temperature of 30 °C. The chromatographic column was an All Chrom NH_2_ (250 mm × 4.6 mm, 5 μm) with a column temperature of 30 °C. The mobile phase was an acetonitrile–water mixture (75:25 *v*/*v*) with an injection volume of 5 μL and a flow rate of 0.8 mL/min. The detector was an evaporative light scattering detector, and the drift tube temperature was 95 °C. The HPLC analysis was carried out by an HPLC column at a flow rate of 0.8 mL/min.

The concentration of lactulose standards was 0.005~1.000 mg/mL. Lactose and galactose followed standard preparation, as above.

### 2.7. Statistical Analysis

All experiments were replicated three times, and the results were expressed as mean ± SD (standard deviation) values. One-way analysis of variance (ANOVA) was used, and Duncan’s multipole test was performed using SPSS Statistics 27 software to determine the significance of each mean (*p* < 0.05).

## 3. Results and Analysis

### 3.1. Screening of the Metabolic Carbon Source Characteristics of Each Lactic Acid Bacteria

After the chemical isomerization synthesis of lactulose, the system contained lactulose, lactose, and galactose. Therefore, microbial fermentation was chosen to reduce the lactose and galactose content and increase the lactulose purity.

A convenient and the most efficiently applied method to determine the growth state of LAB strains is to determine the optical density (OD) spectrophotometrically. The value of the OD_600_ increase indicated that the LAB strain could use lactose and galactose. A direct approach was to estimate the OD at 600 nm (OD_600_) of bacterial cultures in microplates to measure the growth of LAB in different media [16]. Therefore, the LAB medium was used to screen the strains, which could utilize lactose and galactose and not utilize lactulose from 451 strains in our laboratory. The results showed that 75 strains could not use lactulose during the OD_600_ experiments [17]. The 75 strains were preliminarily selected by incubating them in MRS at 37 °C for 16 h with OD_600 (MRS-Glc + Lu) (Δ16 h − 0 h)_ < 0.3 in 451 strains from the laboratory, and the remaining 376 strains with OD_600 (MRS-Glc + Lu) (Δ16 h − 0 h)_ were in the range of 0.3~1.0. After this preliminary selection, 75 strains with OD_600 (MRS-Glc + Lu) (Δ16 h − 0 h)_ < 0.3 were transfected into MRS-Glc + Lu, MRS-Glc + Lac, and MRS-Glc + Gal in culture with 37 °C for 16 h. OD_600_ tested the bacterial fermentation liquid at 0 h and 16 h, respectively.

In total, 11 strains (TM-2-8, DX5-1, Gs23-1, Gs13-2, Gs67-2, Gs33-2, Gs73-1, Gs53-3, DX5-3, Gs54-1, and grx.21) with relatively weak lactulose utilization (OD600 _(MRS-Glc + Lu) (Δ16 h − 0 h)_ < 0.2) and relatively strong galactose utilization (OD600 _(MRS-Glc + Gal) (Δ16 h − 0 h)_ > 0.4) are shown in Figure 1. As shown in Figure 1, the value of OD_600_ for the strain grx.21 was higher than the other strains, which indicated that grx.21 could grow better in galactose medium than the other strains. At this time, the growth in the lactulose medium was lower than the other strain, which means that the lactulose utilization capacity of grx.21 was lower. This was related to the high levels of galactose metabolism genes that were enriched in carbon metabolism. It has also been reported that galactose metabolism is one of the remarkably enriched pathways for genes of *L. rhamnosus* [18]. The value of OD_600_ for strains of Gs23-1 and Gs73-1 was lower than the other strains in the galactose medium, which indicated that the galactose utilization capacity of the strain was lower compared to the other stains. However, the lactulose utilization capacity of the strains Gs23-1 and Gs73-1 was higher than the other strains, as shown in Figure 1. This may be due to the existence of the multiple phosphotransferase system (PTS) for fructose utilization and other complexes of carbon metabolism [19]. Leilei yu et al. (2022) reported diverse carbohydrate-active enzyme-related genes in the strains of *L. curvatus* and proved that all strains of *L. curvatus* utilized galactose and fructose. However, there was also a difference in galactose and lactulose utilization abilities [20].

The value of OD_600_ for strains of Gs53-3 was lower than the other strains in the lactose medium, which means that the lactose utilization capacity was the lowest. However, the lactulose utilization capacity was higher than the strain of grx.21. And the galactose utilization capacity was lower than the strain of grx.21. Therefore, different lactic acid bacteria strains show different utilization of various sugars. These 11 target strains were selected for the following experiments and identification, as shown in Table 1. After sequencing, the 16s rRNA gene sequencing of each strain was carried out using BLAST on the NCBI website. The screening results to achieve the 16s rDNA identification of lactic acid bacteria are shown in Table 1; the strains all presented 16S rRNA gene sequence similarity values higher than 98.5%. When the sequence homologies of 16S rRNA were over 97%, they could be considered to belong to the same species [21]. The strains of DX5-1 and DX5-3 belong to the species of *Lacticaseibacillus paracasei*. The strains of Gs23-1, Gs13-2, Gs67-2, Gs33-2, Gs73-1, Gs53-3, and Gs54-1 belong to the species of *Latilactobacillus curvatus*. The strain of TM-2-8 belong to the species of *Ligilactobacillus salivarius*.

### 3.2. Selection of Lactic Acid Bacterial Strain Utilizing Lactulose, Lactose, and Galactose in MRS Systems

A total of 11 strains were activated and transferred to MRS-Glc + 0.5% Lu + 0.5% Lac + 0.5% Gal medium for fermentation for 48 h. Samples were taken every 8 h, and the remaining amount of different carbon sources during fermentation was detected by HPLC, as shown in Figure 2.

According to the utilization of different sugars, 11 strains can be classified into three groups. The first group is GS23-1, GS33-2, GS73-1, and grx.21, which do not consume lactulose and lactose, as shown in Figure 2a–d. The second group is GS13-2, GS53-3, GS54-1, and GS67-2, as shown in Figure 2e–h, which do not consume lactulose and slowly consume lactose. The third group is DX5-1, DX5-3, and TM-2-8, as shown in Figure 2i–k can slowly consume lactulose and lactose. Among them, TM-2-8 consumed more lactulose and lactose than the other strains and has the ability to utilize more lactulose and less lactose, as shown in Figure 2. The second and third groups with the potential ability to metabolize lactose and grx.21 with high galactose utilization were selected for the subsequent experiments. As shown in Figure 2, the concentration of galactose sharply decreased after fermentation for 16 h by the 11 strains. And the concentration of galactose was close to zero. This indicated that the 11 strains of LAB had higher galactose utilization capacity in the MRS-Glc medium. This was related to galactose, which can enter cells via either permeases encoded by *galP* or phosphotransferase systems encoded by *lacFE* or other unknown PTS systems. It was indicated that LAB strains could internalize galactose via different metabolism pathways so that different strains of LAB for galactose utilization at different rates [22]. The concentration of lactulose and lactose did not significantly change for the stains of GS13-2, GS53-3, GS54-1, and GS67-2 during the fermentation process, with them having low lactose and lactulose utilization capacity, as shown in Figure 2a–d. However, the lactose concentration decreased significantly after fermentation at 8 h for the strain of GS13-2 until 16 h when it leveled off. This indicated that the strain of GS13-2 has a higher lactose utilization capacity. The lactose concentration was slightly decreased during the fermentation process for the strains GS53-3, GS54-1, GS67-2, DX5-1, and DX5-3. Furthermore, the lactose concentration sharply decreased after fermentation for 24 h for the strain of TM-2-8 and was close to zero at 48 h. The LAB strain can reduce the lactose and galactose concentration in the liquid. Therefore, the strains of GS13-2, GS53-3, GS54-1, GS67-2 DX5-1, DX5-3, TM-2-8, and grx.21 were selected to reduce the concentration of lactose and galactose in the chemical synthesis of the lactulose system in the following experiment.

### 3.3. Chemical Synthesis of Lactulose

The prolonged reaction time of lactulose preparation from lactose catalyzed by hot-alkaline-based catalytic isomerization is accompanied by a decrease in pH and the generation of by-products, which reduces the possibility of the intramolecular recombination and isomerization of lactose into lactulose [14].

In accordance with previous studies, the initial reaction pH values were 11 and 12 for the lactose solution used in this study. The lactose solution was heated in a water bath at 60 °C and 70 °C, and the reaction time was 50 min, 70 min, and 90 min, respectively. HPLC was used to analyze the concentration of lactulose in the chemical synthesis system after the hot-alkaline-based catalytic isomerization of lactose into lactulose.

The effect of Initial pH, temperature, and reaction on the lactulose conversion rate and lactulose concentration is shown in Table 2. The lactulose conversion rate is the ratio of lactulose to lactose.

The initial reaction pH was 11 and the reaction temperature was 60 °C and 70 °C. The lactulose conversion rate was 25.9 ± 1.6% at 60 °C for 70 min and 27.2 ± 2.1% at 70 °C for 50 min. The increased reaction temperature could increase the lactulose conversion rate and shorten the reaction time at an initial pH of 11. When the initial reaction pH was 12 and the reaction temperature was 60 °C and 70 °C, the lactulose conversion rates were 29.4 ± 1.3% at 50 min and 31.3 ± 1.2% at 50 min, respectively. The lactulose conversion rates at the initial pH of 12 were significantly higher than at pH 11. Increasing the reaction temperature under the same initial pH conditions could help improve the lactulose conversion rate. Increasing the initial reaction pH could also increase the lactulose conversion rate at the same temperature. Currently, hot-alkaline-based (the alkaline catalyst was NaOH) synthesis presents a 23.9~26.7% yield of lactulose [2]. Therefore, the chemical synthesis of lactulose conditions were chosen as an initial reaction pH of 12 and a reaction temperature of 70 °C for 70 min. The concentration of lactulose was 62.6 ± 2.4 (g/L) at this time in the hot-alkaline-based catalysis system. Then, the hot-alkaline-based catalysis system was cooled to room temperature for the next step of lactobacillus fermentation.

### 3.4. The Concentration of Sugars in the Chemical Isomerization System after Fermentation by LAB

In the system for preparing lactulose through use of the chemical isomerization method, the pH of the solution was 8.5 ± 0.05 after a 70 min reaction at 70 °C with an initial pH of 12. To enable the strains to survive and metabolize the energy substances in the system and the termination of the reaction, the pH of the chemical catalysis system was adjusted to 7 with phosphoric acid. Eight strains of the strains mentioned above were expanded and cultured to collect the logarithmic end-stage of the strains’ bodies, centrifugally washed, and resuspended to the system separately. The concentration of the strains in the chemical isomerization system was 5 g/L.

The solution after inoculation was placed in the incubator at a constant temperature of 37 °C for 48 h. The samples were taken off every 8 h to test the change in the carbon source content of the system through use of HPLC, as shown in Figure 3. The strains were inoculated to the chemical synthesis system of lactulose. The concentration of the galactose, lactose, and lactulose was changed slightly during the fermentation process by the strains of *L. curvatus* GS53-3, GS54-1, GS67-2, *L. pantheris* GS13-2, *L. paracasei* DX5-1, and DX5-3, as shown in Figure 2a–f. This indicated that the utilization capacity of these strains for galactose, lactose, and lactulose is deficient in the chemical catalysis system. The lactose concentration decreased after fermentation for 24 h by the stain of *L. salivarius* TM-2-8, as shown in Figure 3g. This indicated that *L. salivarius* TM-2-8 could utilize lactose in the chemical catalysis system. The concentration of galactose sharply decreased after being fermented by the strain *L. rhamnosus* grx.21, which indicated that the galactose had been used or converted into something else by *L. rhamnosus* grx.21 in the chemical catalysis system. Moreover, the galactose concentration was close to zero after fermentation for 32 h, as shown in Figure 3g. Furthermore, the lactulose concentration was decreased slightly after fermentation for 40 h by the strain *L. rhamnosus* grx.21, as shown in Figure 3g. This indicated that the strains of *L. rhamnosus* grx.21 could grow in the chemical synthesis system of lactulose and reduce the concentration of galactose, which reduced the amount of non-target products in the lactulose synthesis system. The results obtained in this study showed that the strains were subjected to marked metabolic perturbations under environmental stress conditions. As a consequence of stress, strains will lower their metabolic rate to alter their growth and viability [23].

### 3.5. The Effect of Glutamic Acid on the Galactose and Lactose Utilization Capacity of the LAB Strains

In general, glutamic acid promotes the growth of lactic acid bacteria [24,25]. Iliya Dauda Kwoji (2022) reported that glutamic acid has a favorable effect on Lacticaseibacillus [26]. To increase lactose and galactose consumption through LAB strain fermentation in the chemical synthesis system of lactulose, the glutamic acid replaced the phosphoric acid to increase the growth of different strains. The pH of the chemical synthesis system was adjusted to 7 through the addition of glutamic acid before fermentation. This was an attempt to increase the use of non-target production (galactose and lactose), purifying the concentration of lactulose in the synthesis system. The fermentation time was shortened to 18 h according to the growth curve of Lactobacillus. The samples were taken at 0 h, 6 h, 12 h, and 18 h. The change in the sugar content of the system was determined through use of HPLC, as shown in Table 3.

The consumption of galactose was significant in the chemical synthesis system of lactulose for the strains after glutamic acid addition, as shown in Table 3. Regarding the consumption of different sugars, it can be seen that all strains consumed galactose, which was similar to our previous study, as shown in Figure 2. This is also the same compared with the previous report on the ability of LAB to metabolize galactose [22]. It has been reported that almost all LAB strains can metabolize galactose, except some strains of *Streptococcus thermophilus* and *Lactobacillus delbruckii*. This could explain why all strains could consume galactose in this study. The strain of *L. rhamnosus* grx.21 could consume galactose the fastest during the first 6 h of fermentation time, which was significantly different from that of the other strains. Meanwhile, the consumption of lactose and lactulose showed no significant difference when fermented by the strains of *L. paracasei* DX5-1, DX5-3, *L. curvatus* GS53-3, GS54-1, GS67-2, *L. pantheris* GS13-2, and *L. salivarius* TM-2-8 in this study. Therefore, the highest lactose consumption was observed for *L. salivarius* TM-2-8 compared to the other strains.

### 3.6. Effect of Skimmed Milk on Galactose and Lactose Utilization by the LAB Strains

The skimmed milk medium is the most frequently used medium for starter culture production in the dairy industry [24]. Skimmed milk is inexpensive and nutrient-rich, with it containing carbon, nitrogen, and trace elements essential for microbial growth [27]. With the addition of glutamic acid, the strain appeared to be able to utilize the carbon source in the system. However, there was no significant decrease in the lactose content of the system. After adjusting the pH of the system with glutamic acid, sterilized skimmed milk was added to the system to a final concentration of 1.2% (g/100 mL) (considering the lower lactose content and facilitating the downstream process). The fermented samples were taken at 0 h, 6 h, 12 h, and 18 h. The change in the sugar content of the system was determined through use of HPLC, as shown in Figure 4.

All strains appeared to start the ability to utilize galactose and the potential to utilize lactose. Among them, fermentation broth showed a decline, and the lactose concentration was 50.24 ± 2.88 g/L. The galactose concentration decreased slightly but was higher than the other strains’ fermented system, as shown in Figure 4g. The strain of *L. rhamnosus* grx.21 fermentation broth showed a sharp decrease in galactose concentration, with it being close to zero, as shown in Figure 4h. Moreover, the concentration of lactose decreased after being fermented for 12 h by the strain *L. rhamnosus* grx.21, as shown in Figure 4h. The concentration of lactulose decreased slightly during fermentation for 0–12 h and then increased at 18 h of fermentation by the strain *L. rhamnosus* grx.21. This means that the end concentration of lactulose could not be affected by the *L. rhamnosus* grx.21 fermentation process. Though the concentration of galactose and lactose decreased when fermented by the strains *L. curvatus* GS67-2, *L. paracasei* DX5-1, and *L. paracasei* DX5-3, the concentration of target production (lactulose) also decreased during the fermentation time. The strains of *L. curvatus* GS53-3, *L. pantheris* GS13-2, and *L. curvatus* GS54-1 showed the potential to utilize lactose, as shown in Figure 4a–c. However, the concentration of lactose change was not significant during the fermentation time. Therefore, *L. salivarius* TM-2-8 and *L. rhamnosus* grx.21 were selected for the following experiments, which attempted to reduce the concentration of lactose and galactose through the method of continuous fermentation by *L. salivarius* TM-2-8 and grx.21 in the chemical synthesis system.

### 3.7. Preparation of Highly Concentrated Lactulose

Due to the above results, *L. salivarius* TM-2-8 and *L. rhamnosus* grx.21 were mixed for fermentation to reduce the non-target products of lactose and galactose in the chemical synthesis systems of lactulose in this experiment. Therefore, after lactulose preparation through use of the chemical isomerization method, the solution pH was adjusted by using glutamic acid to terminate the reaction with glutamic acid and skimmed milk (1.2%). Then, the strain of *L. salivarius* TM-2-8 was added to ferment for 18 h to reduce the concentration of lactose, and *L. rhamnosus* grx.21 was added after adjusting the pH to 7 with NaOH to continue fermentation from 18 h to 36 h to reduce the concentration of galactose. The concentration of LAB was 5 g/L. The samples were taken off every 6 h. The concentrations of lactose and lactulose were determined through use of HPLC, as shown in Figure 5.

The lactose concentration was slightly increased at fermentation for 6 h, which was related to the skimmed milk’s addition. Then, the lactose concentration began to reduce after fermentation for 12 h to 18 h of fermentation by the strain *L. salivarius* TM-2-8. Then, the concentration of lactose continued to decline after 18 h to 36 h of fermentation by *L. rhamnosus* grx.21. Furthermore, the concentration of lactulose began to decrease after 18 h, which indicated that continuous fermentation by *L. rhamnosus* grx.21 could reduce the content of target production—lactulose in the mixture system. The lactulose concentrations also decreased during the mixed *L. salivarius* TM-2-8 and *L. rhamnosus* grx.21 fermentation process. During the fermentation process, the concentration of lactulose (59.73 ± 1.8 g/L) was significantly higher than lactose (48.96 ± 2.92 g/L) at 18 h. The fermentation broth was taken, autoclaved, and kept at −4 °C for the next experiment in the highest concentration of lactulose at 18 h. The enzyme catalysis process used the single strain of the *L. salivarius* TM-2-8 fermented system followed by a chemical synthesis system.

### 3.8. Enzymatic Synthesis Increases Lactulose Synthesis

β-galactosidase has been used to catalyze lactulose synthesis. The hydrolytic activity of β-galactosidase hydrolyzes lactose to galactose and glucose and transfers galactose to the fructose receptor to produce lactulose through the transglycosylation activity of β-galactosidase. Because galactose and lactose are contained in the microbial fermentation system of the hot-alkaline-based catalytic system, β-galactosidase was used in this study to catalyze the increase in lactulose content further.

Two sets of fermentation broths were taken, their pH was adjusted to 7, and different concentrations of β-galactosidase (0.5 U/mL, 1 U/mL, 2 U/mL, 3 U/mL, and 4 U/mL) were added to the fermentation broths and reacted at 50 °C for 18 h. At the same time, fructose was added in a 2:1 (lactose: fructose w: w) mass ratio. Samples were taken at 0 h, 3 h, 6 h, 12 h, and 18 h to test the concentration of lactulose through use of HPLC, as shown in Figure 6a.

When the β-galactosidase was added at 1 and 2 U/mL, the concentration of lactulose changed slightly, which means that the enzyme catalysis concentration was lower than the catalytic substrate. The concentration of lactulose was significantly higher in the fermentation broth with the β-galactosidase addition of 3 U/mL with catalysis for 3 h. When the enzyme catalysis time increased, the concentration of lactulose slightly decreased. This was because lactulose synthesis is kinetically controlled, meaning that after a peak concentration, β-galactosidase tends to hydrolyze the produced lactulose [28]. The concentration of lactulose decreased with β-galactosidase addition of 4 U/mL. This was due to the fact that an excessive increase in the enzyme concentration would increase the rate of lactulose hydrolysis at the same time. This then lowers the lactulose yield [29].

As shown in Figure 6b, the lactose concentration decreased with the β-galactosidase catalysis process. When the enzyme concentration addition was 4 U/mL, the lactose concentration was lowest at 3 h. Therefore, the maximum concentration of lactulose (74.89 ± 1.68 g/L) and the minimum concentration of lactose (31.25 ± 3.2 g/L) were achieved with an enzymatic concentration of 3 U/mL for 3 h.

## 4. Conclusions

The strains of *L. rhamnosus* grx.21, *L. pantheris* GS13-2, *L. curvatus* GS53-3, *L. curvatus* GS54-1, *L. curvatus* GS67-2, *L. paracasei* DX5-1, *L. paracasei* DX5-3, and *L. salivarius* TM-2-8 showed weak utilization of lactulose selected from 451 strains in the laboratory strain library. These strains were used to ferment the chemical synthesis system of lactulose to reduce the lactose and galactose content. Then, β-galactosidase was added to further catalyze the lactose synthesis of lactulose, which increased the concentration of lactulose.

In this study, the optimal conditions for lactulose production through chemical isomerization were determined through testing at 70 °C for 50 min, and the highest concentration of lactulose was 31.3 ± 1.2%. Then, the higher utilization of lactose and galactose and weaker lactulose utilization strains were screened using different sugar mediums from 451 strains in the laboratory strain library. The glutamic acid and skimmed milk were also added to investigate lactose, galactose, and lactulose utilization. It was found that the glutamic acid and skimmed milk addition were able to increase the lactose metabolism of *L. salivarius* TM-2-8, which meant that the lactose concentration decreased. The concentration of lactose and galactose also decreased when fermented by the strain *L. rhamnosus* grx.21 in the chemical synthesis system of lactulose. Then, the strains of *L. salivarius* TM-2-8 and *L. rhamnosus* grx.21 were decided for continuous fermentation to reduce the lactose and galactose concentrations in the chemical synthesis system of lactulose. It was found that the single strain of *L. salivarius* TM-2-8 when fermented has better efficiency than the two strains’ fermented system. Therefore, the single strain of the *L. salivarius* TM-2-8 fermented system at 18 h was selected to participate in the following enzymatic reaction. After fermentation, the lactose concentration was 48.96 ± 2.92 g/L and the concentration of lactulose was 59.73 ± 1.8 g/L. β-galactosidase was used to catalyze lactulose synthesis to increase the concentration of lactulose. It was found that the β-galactosidase catalysis conditions were 50 °C for 3 h with an enzymatic concentration of 3 U/mL. Under these conditions, a higher lactulose concentration of 74.89 ± 1.68 g/L was obtained. The concentration of lactose was 31.25 ± 3.2 g/L. Therefore, hot-alkaline-based catalysis lactose isomerization to lactulose followed by *Lactobacillus* fermentation could reduce the concentration of lactose and galactose. Then, the β-galactosidase catalysis process could increase the concentration of lactulose and decrease the concentration of lactose. All in all, chemical synthesis combined with lactic acid bacteria fermentation and enzyme catalysis has a potential application in purifying and increasing lactulose concentration.

## Figures and Tables

**Figure 1 foods-12-04317-f001:**
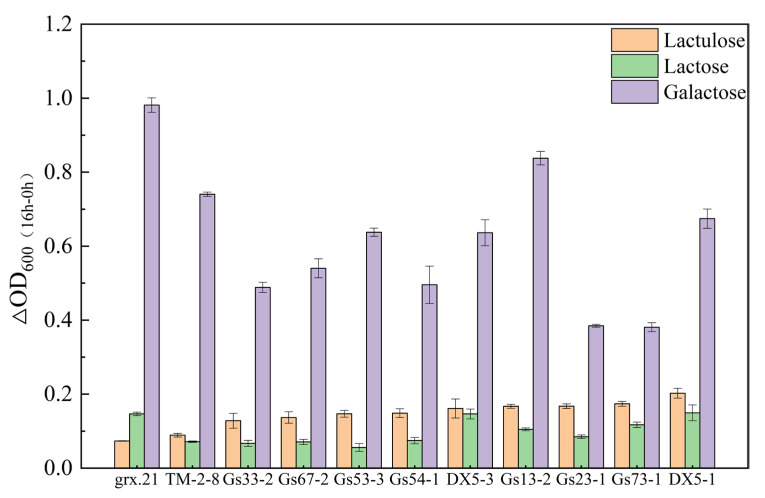
Growth of 11 strains in MRS-Glc + Lu, MRS-Glc + Lac, and MRS-Glc + Gal medium.

**Figure 2 foods-12-04317-f002:**
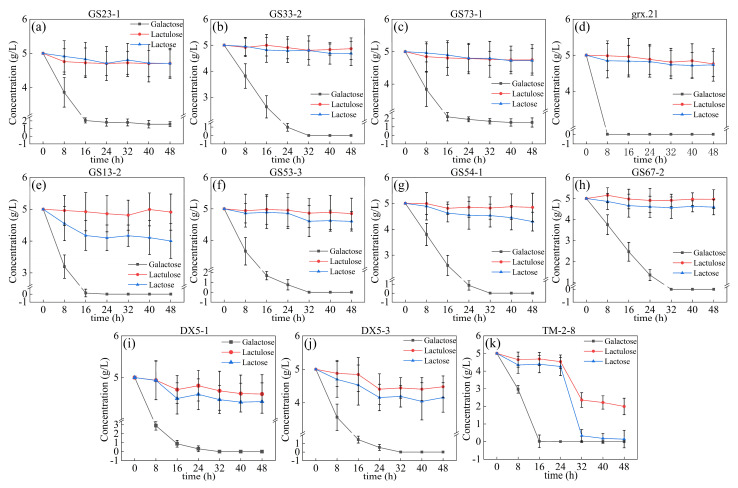
The concentration of lactulose, lactose and galactose in the MRS system after fermentation by 11 strains: GS23-1 (**a**), GS33-2 (**b**), GS73-1 (**c**), grx.21 (**d**), GS13-2 (**e**), GS53-3 (**f**), GS54-1 (**g**), GS67-2 (**h**), DX5-1 (**i**), DX5-3 (**j**), and TM-2-8 (**k**).

**Figure 3 foods-12-04317-f003:**
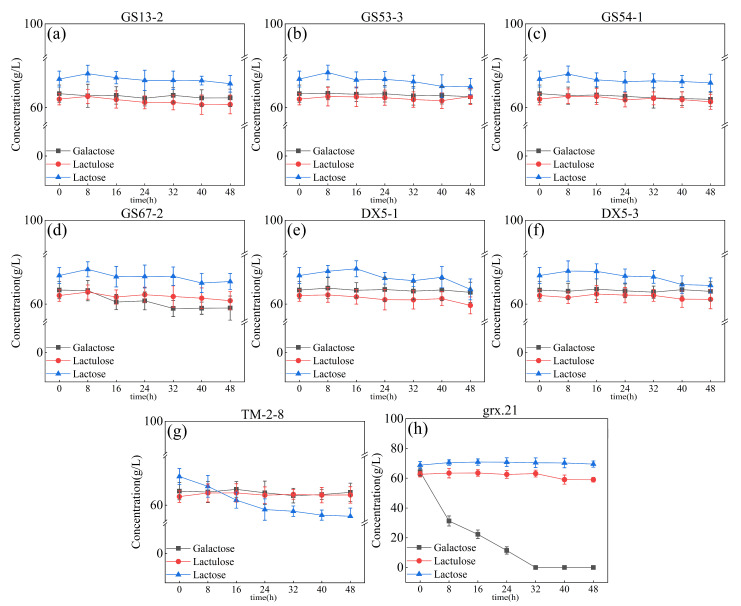
The concentration of galactose, lactulose, and lactose fermented by different strains in the chemically synthesized lactulose systems: GS13-2 (**a**), GS53-3 (**b**), GS54-1 (**c**), GS67-2 (**d**), DX5-1 (**e**), DX5-3 (**f**), TM-2-8 (**g**), and grx.21 (**h**).

**Figure 4 foods-12-04317-f004:**
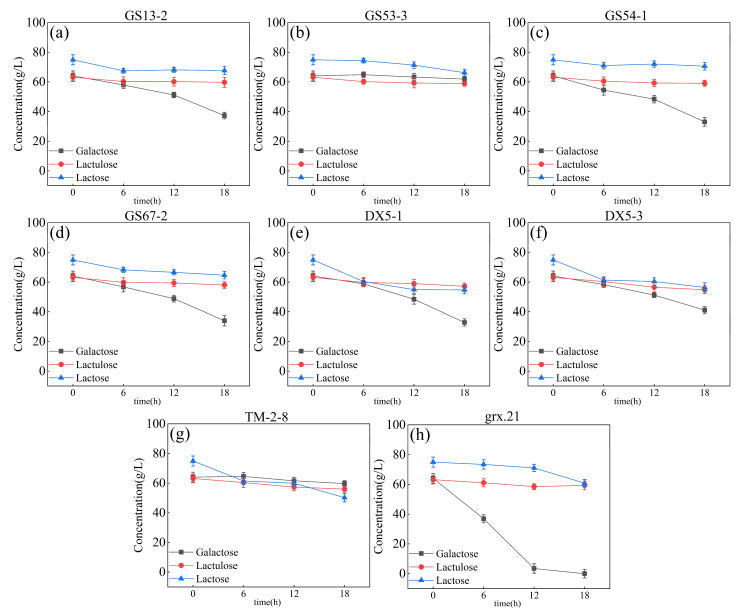
Effect of skim milk on the galactose, lactulose, and lactose utilization characteristics of strains: GS13-2 (**a**), GS53-3 (**b**), GS54-1 (**c**), GS67-2 (**d**), DX5-1 (**e**), DX5-3 (**f**), TM-2-8 (**g**), and grx.21 (**h**).

**Figure 5 foods-12-04317-f005:**
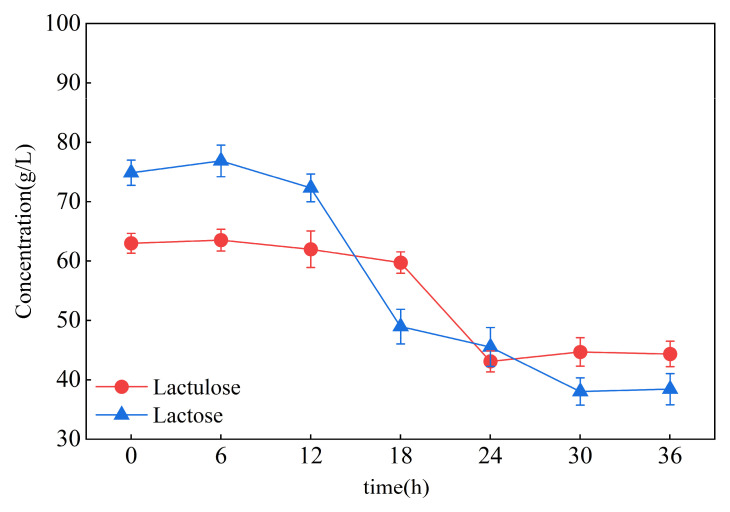
Chemically synthesized systems of lactulose fermented by LAB.

**Figure 6 foods-12-04317-f006:**
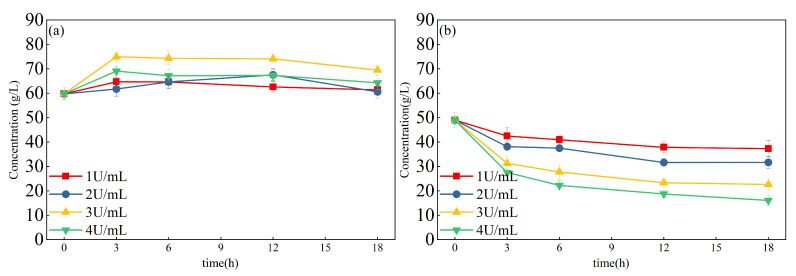
Effect of enzyme concentration and reaction time on lactulose concentration. (**a**) shows the concentration of lactulose (g/L). (**b**) shows the concentration of lactose (g/L).

**Table 1 foods-12-04317-t001:** Homology comparison results of 16s rDNA gene sequences of 11 strains.

Species	Strains
*Ligilactobacillus salivarius*	TM-2-8
*Lacticaseibacillus* *paracasei*	DX5-1
*Lacticaseibacillus paracasei*	DX5-3
*Latilactobacillus curvatus*	GS23-1
*Latilactobacillus curvatus*	GS67-2
*Latilactobacillus curvatus*	GS33-2
*Latilactobacillus curvatus*	GS73-1
*Latilactobacillus curvatus*	GS53-3
*Latilactobacillus curvatus*	GS54-1
*Lacticaseibacillus pantheris*	GS13-2
*Lactobacillus rhamnosus*	grx.21

**Table 2 foods-12-04317-t002:** Maximum conversion of lactulose and the reaction time required to reach maximum conversion under different conditions in the chemical synthesis system of lactulose.

pH	Temperature (°C)	Maximum Concentration of Lactulose (g/L)	Maximum Conversion (%) (Lu:Lac *w*:*w*)	Maximum Conversion of Reaction Time (min)
11	60	51.8 ± 3.1	25.9 ± 1.6% ^d^	70
12	60	58.8 ± 2.6	29.4 ± 1.3% ^b^	50
11	70	54.4 ± 4.2	27.2 ± 2.1% ^c^	50
12	70	62.6 ± 2.4	31.3 ± 1.2% ^a^	50

^a, b, c, d^ statistically significant differences in the maximum conversion of lactulose (*p* < 0.05).

**Table 3 foods-12-04317-t003:** Effect of glutamic acid on the concentration of galactose and lactose in the chemically synthesized systems of lactulose fermented by different strains.

Time	Sugars	The Content of Sugars of Different LAB Strains at Different Time (g/L)							
		Gs13-2	Gs53-3	Gs54-1	Gs67-2	DX5-1	DX5-3	Tm-2-8	grx.21
0 h	Galactose	63.98 ± 3.22	-	-	-	-	-	-	-
	Lactulose	62.82 ± 3.62	-	-	-	-	-	-	-
	Lactose	69.13 ± 2.37	-	-	-	-	-	-	-
6 h	Galactose	48.77 ± 3.42 ^d^	54.99 ± 1.55 ^c^	46.04 ± 2.78 ^g^	48.21 ± 3.78 ^e^	47.90 ± 3.30 ^f^	58.82 ± 1.89 ^b^	59.74 ± 3.54 ^a^	42.28 ± 1.92 ^h^
	Lactulose	63.65 ± 2.00 ^c^	63.05 ± 2.80 ^f^	62.37 ± 2.14 ^h^	63.71 ± 2.88 ^b^	62.94 ± 1.60 ^g^	63.57 ± 2.17 ^d^	63.24 ± 2.81 ^e^	63.72 ± 2.90 ^a^
	Lactose	67.10 ± 1.85 ^f^	67.29 ± 2.42 ^e^	68.05 ± 2.79 ^a^	67.70 ± 2.43 ^b^	67.60 ± 2.83 ^c^	67.03 ± 1.55 ^h^	67.30 ± 1.54 ^d^	67.08 ± 2.49 ^g^
12 h	Galactose	42.90 ± 1.51 ^e^	50.28 ± 3.24 ^c^	47.25 ± 2.92 ^d^	40.02 ± 2.76 ^f^	38.69 ± 1.67 ^g^	56.87 ± 2.11 ^b^	57.49 ± 2.70 ^a^	5.10 ± 2.68 ^h^
	Lactulose	64.11 ± 3.75 ^a^	64.09 ± 1.70 ^b^	62.37 ± 3.72 ^g^	62.31 ± 2.18 ^h^	63.38 ± 3.11 ^c^	62.96 ± 2.35 ^f^	63.09 ± 1.94 ^e^	63.17 ± 2.43 ^d^
	Lactose	67.12 ± 2.61 ^b^	67.06 ± 1.98 ^c^	65.93 ± 1.97 ^e^	64.67 ± 3.17 ^g^	66.61 ± 3.70 ^d^	65.85 ± 3.76 ^f^	63.97 ± 2.71 ^h^	67.43 ± 2.90 ^a^
18 h	Galactose	35.18 ± 2.26 ^d^	40.22 ± 2.19 ^c^	33.68 ± 3.62 ^e^	31.87 ± 2.24 ^g^	33.25 ± 3.04 ^f^	56.13 ± 2.14 ^a^	55.56 ± 3.71 ^b^	0 ^h^
	Lactulose	63.65 ± 1.75 ^b^	64.01 ± 2.62 ^a^	62.15 ± 1.93 ^e^	60.08 ± 2.70 ^h^	62.86 ± 2.53 ^d^	61.66 ± 2.19 ^g^	61.80 ± 2.66 ^f^	63.48 ± 3.24 ^c^
	Lactose	66.62 ± 1.94 ^b^	66.80 ± 3.50 ^a^	64.52 ± 1.75 ^e^	63.04 ± 1.67 ^g^	65.87 ± 3.57 ^d^	64.00 ± 2.76 ^f^	62.62 ± 2.10 ^h^	66.09 ± 2.93 ^c^

Mean values were compared within a row. The same letter means no significant difference (*p* > 0.05), and different letters mean a significant difference (*p* < 0.05).

## Data Availability

The data presented in this study are available from the authors upon request.

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
