# Peer review of "Increase in Lactulose Content in a Hot-Alkaline-Based System through Fermentation with a Selected Lactic Acid Bacteria Strain Followed by the β-Galactosidase Catalysis Process"

_foods, 2023, doi:10.3390/foods12234317_

Round 1

Reviewer 1 Report

Comments and Suggestions for Authors The comments are included in the attached pdf file.

Ícono de validado por la comunidad

Comments on the Quality of English Language The comments are included in the attached pdf file.

Ícono de validado por la comunidad Ícono de validado por la comunidad

Reviewer 2 Report

Comments and Suggestions for Authors

The manuscript is difficult to read because, in the first place, English needs strong revision. Second, I do not understand the final objectives of the study. As it is a more chemical study than microbiological, I will detail only some microbiological observations, which should be added in a review that someone should do with chemical formation:

- The genus and species of microorganisms must always be written in italics. Review References and all text

- When they express %, the units should be clarified (gr for every 100 gr of what?)

- "Lactobacilli" is not written in Italics. It is not Latin, it is a plural in English.

- make clear the final objectives of the study and the usefulness of the results

Comments on the Quality of English Language

You need deep review to properly read the text

Reviewer 3 Report

Comments and Suggestions for Authors

This work combines LAB fermentation with biocatalysis to increase and purify lactulose production. It is interesting, however I consider thay the way or results presentation must be improved.

I suggest to transform Figure 2 and 3 in bars graph with the % of consumed sugar at the end of fermentation

The Figure 5 can be transformed in a table, again in bars graph with the % of consumed sugar

According to the text in lines 349-352, the addition of glutamic acid was not effective as expected, thus I suggest to send the figure 5 as supplementary material

Round 2

Reviewer 2 Report

Comments and Suggestions for Authors

In my opinion, the manuscript appears improved and could be accepted if the chemical reviewers agree. There are a few things left to fix: - line 531: "subsp." it is not written in italics - line 496: replace "Lactobacillus" with "lactobacilli"           ​

Author Response

Answer: According to the reviewer’s advice, the “subsp” has been revised in line 523, and the "Lactobacillus" replaced "lactobacilli" in line 487.

Reviewer 3 Report

Comments and Suggestions for Authors

The manuscript was improved

Author Response

Dear reviewer,

Thanks for your comments and suggestions again.

Best regards.